# Development and Evaluation of Cultural Competence Course on Undergraduate Nursing Students in Vietnam

**DOI:** 10.3390/ijerph19020888

**Published:** 2022-01-13

**Authors:** Trang-Thi-Thuy Ho, Jina Oh

**Affiliations:** 1Faculty of Nursing, Hue University of Medicine and Pharmacy, Hue University, Hue 49120, Vietnam; htttrang.humed@hueuni.edu.vn; 2Institute of Health Science, College of Nursing, Inje University, Busan 47720, Korea

**Keywords:** cultural competence, nursing education, nursing student, Vietnam

## Abstract

Cultural competence is a crucial requirement of nursing to promote caring for patients with diverse backgrounds. The purpose of this study was to develop a cultural competence course and to evaluate the effects of the course on undergraduate nursing students in Vietnam. A concurrent triangulation mixed-methods study was adopted using quantitative and qualitative data sources. Sixty-six nursing students were recruited for the following groups: cultural competence course with field experience (*n* = 22), stand-alone cultural competence course (*n* = 22), and a control group (*n* = 22). The findings indicated that significant group by time interactions in total cultural competence score (F = 66.73, *p* < 0.001) were found. Participants’ perceptions reflected on three categories: (a) journey to cultural competence, (b) satisfaction of cultural competence course, and (c) suggestions for improvements. No statistically significant differences between the two experimental groups were revealed, but “obtaining cultural experiences” and “expanding understanding of cultural competence through field experience” were immersed from participants having field experience. It is vital to expand cultural competency education into nursing curricula to enhance nursing students’ perspective of culturally competent care.

## 1. Introduction

Cultural competence has become an essential component and requirement in nursing education to reinforce the need for culturally competent care. Nursing education organizations have for a long time recommended that cultural competence should be incorporated into the nursing curriculum [1,2,3]. Preparation of students on cultural competence through nursing education has contributed significantly to the quality of professional nursing practice [4,5,6]. Improving individual cultural competency development is a crucial educational achievement utilizing different approaches of training [2,3,7]. Comprehensive education and training are key to improving cultural competence in nursing care [8,9].

With 54 ethnic groups in Vietnam, cultural diversity is a visible aspect of the challenges faced by society [10] in the context of health inequity and disparity [11,12]. This diversity emphasizes the pressing need to develop cultural competence for Vietnamese nurses in order to achieve health equality for all patients, irrespective of their cultural background [13]. However, insufficient understanding of cultural competency has been revealed among Vietnamese nursing students [14]. Furthermore, cultural competence appears to be an unfamiliar term in nursing education in Vietnam, and it is not yet considered a topic in the nursing curricula; hence, current cultural education content is insufficient to achieve the goals of nursing students in Vietnam.

Cultural competence education is recognized as a broad subject, with a variety of current teaching and learning strategies and different training standards [15]. Educators have been asked to carefully consider the content and delivery methods for the introduction of cultural competency topics into the nursing curricula [16], as it is essential to offer culturally appropriate education to nursing students [7]. In this study, application of the ADDIE model guideline offers a flexible and systematic [17,18] approach for developing appropriate cultural education in the Vietnamese context. Thus, the purpose of this study is to develop a cultural competence course and to explore the effects of a cultural intervention to improve the cultural competence of nursing students in Vietnam to meet the needs of culturally competent care.

## 2. Materials and Methods

### 2.1. Development of the Cultural Competence Course

The course was developed following the systematic approach of the ADDIE model [17,18] following five phases: analysis, design, development, implementation, and evaluation. In the pre planning phase, multiple resources were analyzed to create instructional goals and learning objectives. The design phase facilitated student’s learning and interaction with the materials [18,19]. Based on results of the first two phases, the authors developed and created a factual sample for the instruction design and course materials [20]. Finally, the main activities were transformed and evaluated in the implementation and evaluation phases [19,20].

In the analysis stage, a systematic review of previous training programs on cultural competence from 2005 to 2018 was conducted. The purpose of analysis was to propose the general course construction, concurrent with a preliminary needs assessment with nursing educators and students involving educational requirements on cultural competence, educational environment, and applicable cultural content. The nursing educators’ perspective was interpreted using a questionnaire and in-depth interviews. Nursing students’ perception of cultural competence, educational environment, and potential content was performed utilizing multiple tools and focus group interviews. The findings indicate that the nursing curriculum has not routinely focused on cultural competence, and previous students’ foundational knowledge of cultural competence was limited. Similarly, previous studies demonstrated a need to enhance cultural competence for nursing students [3,13,14]. Thus, these course goals began with the fundamentals of cultural competence.

In the development and design stage, the cultural competence course adopted the framework of the Campinha-Bacote model of cultural competence [21]. This model identified that cultural competence was as ongoing process in which health care providers aimed to effectively work within the cultural context of patient. The author described cultural competence involving five constructs: cultural knowledge, awareness, skill, sensitivity, and encounters. The constructs were (a) lecture of each concept, (b) illustrating the aspects of the educational foundation of cultural diversity, (c) learners’ self-examination of cultural and professional background, (d) ability to respect and appreciate patient beliefs, valuing of culture, and health behavior; and (e) understanding of communication strategy and cultural assessment. In addition, cultural encounters enriched the participants’ learning through applying field experience.

Subsequently, the course consisted of one credit for 10 h of class lecture, 4 h for practice, and 1 h for orientation and a final examination over 7 weeks. The course content was developed systematically based on the results of the analysis phase and theoretical framework. The course outcome covered broad topics such as: describe definitions of culture and cultural competence; identify concept, theories, and models related to cultural competence; discuss the impact of culture on culturally competent nursing care; perform culturally competent nursing assessments in diverse settings; and discuss nursing strategies to promote culturally competent care. Four units were classified with culture, cultural diversity, cultural nursing competence, and cultural nursing competence in the Vietnamese context (Table 1). Gagné’s nine events of instruction [22] were applied to ensure that the course was effective.

A lecture about cultural competence with similar content was introduced to the two experimental groups. The two teaching approaches of discussion and field experience were divided into different practical settings and contributed to the experimental groups separately. The first experimental group attending a cultural competence course with field experience had the chance to practice in a minority community in Vietnam. The first group of students interacted with diverse patients directly, communicated with them, and performed the Purnell model of cultural assessment [23]. Collaboration with local health care providers using guest speakers and home visits occurred during the field experience. The students also took part in a case study conference to explore their own experience during the fieldwork. The second experimental group took the stand-alone cultural competence course, which presented and discussed selective topics in a classroom setting.

The assessment approaches utilized were a quiz, reflective writing, and a group presentation. A ubiquitous learning environment was adopted to manage the course and support teaching and learning activities. To ensure course validity, the course blueprint and design were assessed by four nursing experts from Vietnam and South Korea and revised to form the cultural competence course.

### 2.2. Study Design and Participants of Cultural Competence Course

As an intervention and evaluation stage, a concurrent triangulation mixed-methods design was used in this study, with a quasi-experimental longitudinal design. Data were collected at the baseline of four weeks before intervention (T1), pre-test before intervention (T2), immediate post-test after intervention (T3), and at an eight-week follow-up (T4) to evaluate the effects of the intervention (Figure 1). Because all students participated in clinical practicum during the first four weeks, it was necessary to check the effects of clinical experience on cultural competence before intervention. A focus group interview was conducted upon completion of the cultural competence course.

The 66 participants were third-year nursing students recruited from Hue University of Medicine and Pharmacy, Vietnam. The students were allocated to three groups after getting agreement of attendance spontaneously in this study. Sample size was estimated using the G*Power analysis program for repeated-measures ANOVA with effect size = 0.33, α err prob = 0.05, correlation of 0.50, and power = 0.80. To compensate for dropouts, the 66 students who participated were divided into three groups: cultural competence course with field experience (experimental group I); stand-alone cultural competence course (experimental group II); and non-participation in the cultural course (control group). Of the 66 students, four students did not complete the questionnaire on the homogeneity test and did not attend intervention in the experimental groups, and two students refused to participate in the control group. The final sample consisted of 60 students, including 40 students for the two experimental groups and 20 students for the control group.

### 2.3. Instruments for Course Evaluation

Qualitative and quantitative data analyses were conducted in the evaluation process based on the guideline of Kirkpatrick’s evaluation model including learning and reaction levels [24].

#### 2.3.1. Quantitative Measures

Regarding learning level from Kirkpatrick’s first evaluation model, the Nurse Cultural Competence Scale [NCCS] [25] was used to identify the cultural competence level. The tool includes four domains of cultural awareness, cultural knowledge, cultural sensitivity, and cultural skill, with the reliability of subscales ranging from 0.78 to 0.96. In this study, Cronbach’s alpha of the overall scale was 0.85, and ranged from 0.69 to 0.90 in the subscales, indicating high internal consistency.

A student satisfaction survey [26] assessed the satisfaction of training in the reaction level of Kirkpatrick’s evaluation model using three domains: teaching, assessment, and generic skills and learning experiences. Cronbach’s alpha scores in the original article ranged from 0.77 to 0.82. In the present study, Cronbach’s alpha value for the total was 0.92; subscale values ranged from 0.78 to 0.89.

The instruments were translated by the researcher from English into Vietnamese using the WHO process of translation [27] and revised using the comments from two expert panels. The back-translation phase was conducted by a fluent bilingual consultant. The original and back-translation versions were compared and modified to increase the quality of translation. Additionally, forward-backward translation was used in the qualitative data.

#### 2.3.2. Qualitative Measures

Four focus group interviews were sufficient to explore the effects of the cultural competence course. Open-closed questions were used in the learning level to understand participants’ perception of their cultural competence, comprising the following: Please tell me your experience to attend the cultural competence course? How does the course increase your awareness/knowledge/skill of cultural competence? How were you able to meet the objectives? What else do you want to know about cultural competence? Do you have any other comments about your experience on this course?

Focus group questions evaluating satisfaction included: Tell me about your cultural competence course satisfaction (content, material, instructors, facilities)? What teaching approaches/material are the most effective/ineffective? What support might you need to apply what you learned?

### 2.4. Ethical Considerations

To protect the rights of participants, the study was conducted in accordance with Declaration of Helsinki and the protocol was approved by two universities (H2019/002; 2019-06-017-003). A written consent form was obtained if students agreed to participate once the purpose of the study, procedures, and confidentiality had been clearly explained. As a motivational incentive, each student received a gift card during their participation in the experiment.

### 2.5. Data Collection

The study was conducted from September 2019 to May 2020. The data collection process was approved by two institutional review boards of the affiliated institutions. Multiple data sources were collected before and after the intervention. To ensure the anonymity of the survey, participants were not requested to provide their names, only their date of birth to match the survey at four time points. Participants were not informed that there were three groups and were asked not to communicate with their peers any information about the training.

### 2.6. Data Analysis

This study used SPSS 26.0 software (SPSS Inc., Chicago, USA). Descriptive statistics were used to describe the characteristics of the participants. Chi-square test, ANOVA, paired *t*-test, and repeated measures analysis of variance were applied to compare data.

ATLAS.ti version 8.0 was utilized to analyze qualitative data using content analysis. Using the qualitative content analysis process of Elo and Kyngäs [28] containing three main phases of preparation, organizing, and reporting. The combination of different methods was used in this study. Interview, observation, and field notes were involved for the preparation phase; the researchers analyzed data and performed careful follow up on the entire analysis process and categorization for the organization phase; and the categorization process was explained by using tables and quotations for reporting phase. Additionally, the guidelines of Elo and colleagues [29] was exploited in order to improving trustworthiness of this study.

## 3. Results

### 3.1. Quantitative Findings of Effects of Cultural Competence Course 

#### 3.1.1. Baseline Characteristics of Students

The average age of the students was 20 years old, and all the students were female. Most reported not following a religion. All students had not attended any courses or subjects related to cultural competence. Eleven students had taken care of patients in a minor race or ethnic group, and 21 students had lived in an environment of people with diverse race/ethnicities. The study found that the three groups did not significantly differ in baseline characteristics. There were no statistically significant differences among the three groups in total NCCS score and subscales in the homogeneity test (*p* > 0.05) (Table 2).

Moreover, there was no statistically significant differences among the three groups in total NCCS score and subscales at four weeks before intervention and immediately before intervention (*p* > 0.05) (Table 3).

#### 3.1.2. Effectiveness of Cultural Competence Course among Nursing Students

A comparison of nursing student cultural competence in the developed course at pre- and post-test revealed a significant improvement. Using repeated-measures ANOVA with the Greenhouse Geisser correction, the results showed significant interactions between groups by time (from baseline to immediately, and eight weeks after the program) in total NCCS scores with significant time (F = 66.73, *p* < 0.001), interaction of time with group (F = 15.99, *p* < 0.001), and groups (F = 7.59, *p* = 0.001). Additionally, the findings for the subscales showed significant interactions among the three groups by time, in cultural awareness (F = 2.75, *p* = 0.017), cultural knowledge (F = 15.51, *p* < 0.001), cultural sensitivity (F = 3.93, *p* = 0.002), and cultural skill (F = 17.13, *p* < 0.001) (Table 4).

Post hoc tests using the Bonferroni correction revealed significant differences in total NCCS scores between the control group and experimental group I (*p* = 0.001) and experimental group II. In the cultural knowledge subscale, there was a significant difference between the control group and experimental group I, and no significant difference between the control group and experimental group II. In cultural skill, there was a significant difference between the control group and experimental group I (*p* = 0.001) and experimental group II. In subscales of cultural awareness and sensitivity, there was no difference between the control group and experimental groups I and II. There was no statistically significant difference in total NCCS score and subscales between experimental group I and experimental group II (*p* > 0.05).

After Bonferroni adjustment, a statistically significant difference was found between the mean scores of total NCCS scores before intervention (baseline and pre-test), and after intervention (immediate post-test and at eight-week follow-up), suggesting that the cultural competence program was observably effective in increasing participants’ cultural competence both immediately and over time. However, mean scores at the follow-up decreased continuously compared to the immediate post-test scores, although the differences were not significant.

#### 3.1.3. Level of Student Satisfaction

The satisfaction levels among students who attended the cultural competence course were examined to determine the highest and lowest student satisfaction rate with each domain. The highest student satisfaction rate related to the teaching domain (4.62 ± 0.30), and the lowest student satisfaction rate related to the assessment domain (3.98 ± 0.46). The overall mean student satisfaction level after attending the cultural competence course was calculated to be 4.40 using a 5-point Likert scale. There was no significant difference in total student satisfaction between the experimental groups (*p* > 0.05) (Table 5).

### 3.2. Qualitative Results of Effects of Cultural Competence Course

A total of 24 participants attended the cultural competence course. Their responses produced the nursing student perspective on the cultural competence course and particularly their reflections on their field experience (Table 6).

#### 3.2.1. Nursing Student’s Perspective on Cultural Competence Course

From the student responses, three categories were highlighted: (a) journey to cultural competence, (b) satisfaction of cultural competence course, and (c) suggestion for improvements.

Concerning the journey to cultural competence, participants recognized that cultural competence was a continuous process (Figure 2). They had begun the journey to discover cultural competence through gaining cultural knowledge, developing their cultural skills, and encouraging their attitude through the five constructs of cultural knowledge, awareness, skill, sensitivity, and encounters. This course was introduced as the fundamentals of cultural competence.

This course provided sufficient understanding of cultural knowledge, particularly of the culture and health issues of Vietnamese ethnic groups. Participants knew how to build trust in their relationships with patients and how to utilize patient information through forms of communication, enhancing communication skills during interaction with patients, and using family caregivers and neighbors as interpreters for effective communication. Students recognized that the Purnell model was an appropriate model for cultural assessment in diverse patients. They all perceived that the course made them change their own thinking about people from a different culture. They were aware of the importance of empathy; felt empathy and respected patients’ cultural beliefs; and accepted patients’ health beliefs and health behaviors. Others commented that they felt confident in communicating with people from a different culture. “*I thought I was a bit discriminating against them* (LGBTQ+ group). *I saw them as different people*, *I didn’t think they were good people…After studying*, *I gradually accepted them*, *and now I feel they are normal* (IDSEG3)”; “*I didn’t know Cotu and Van Kieu ethnic groups before*, *it made me afraid, but after talking with them*, *they have good points, friendly, and kindness*. *It’s not strange to me. I should respect their cultures” (IDSEG6).*

Subsequently, the effects of this educational training promoted the participant’s motivation based on their needs, goals, and expectations in the process of becoming culturally competent. One stated “*I think there have been only a few sessions and field trips where I have gained so much knowledge. I was satisfied when I joined the course. After that, based on the models and background knowledge, I can build my knowledge* (IDSEG1)”.

Moreover, participants stated that they were pleased overall with the cultural competence course. Some said that the course had exceeded their expectations. “*I feel I exceeded my goal in this course. I achieved more than I expected* (IDSEG2).” Participants were satisfied with the content of the course, which was provided appropriately with students’ achievement. They felt interested though each lesson and gained more new knowledge. Multiple teaching and learning strategies helped them to explore cultural competence. However, they had difficulty in reflective writing because it was an unfamiliar method. The course was regarded as well-prepared, with a convenient ubiquitous learning system and a comfortable educational environment. The students felt they could speak their opinion and felt free to ask and answer questions. “*The content of course is enough suitable. I learn a lot of knowledge. After taking this course, I fully understand and could apply it* (IDSEG8)”.

Regarding suggestions for improvements, participants’ suggestions focused on the content of the course and the teaching and learning strategy. Course content should include more lessons about Vietnamese and foreign cultures. In addition, some expressed that they had difficulty in understanding the models related to cultural competence. The use of video offered a detailed explanation of abstract concepts and models and should be applied in case studies and role-play. “*It should add some common foreign culture, because I interact not only with ethnic minorities but also with foreigners* (IDSEG4)”.

#### 3.2.2. Nursing Students’ Reflection of Field Experience

Two categories emerged to describe the students’ impressions of field experience: (a) obtain cultural experience, and (b) expand understanding of cultural competence through field experience.

Participants recognized field experience as an opportunity for an in-depth understanding of local culture. Local culture was unfamiliar at first, and this impressed the students. They could discover and learn local culture anytime and anywhere and recognize the importance of language and understand its barriers. They dealt with some issues through learning simple words of greeting from local health care providers and using these words to promote easy communication. “*Even in rest periods, we learned a lot about their culture. When we go to a market, we know about their food…or even we play with the children, and we know more about their culture”* (IDSEG2).

Concerning understanding cultural competence through field experience, participants indicated that they gained an understanding of culture as well as cultural competence through practice in the field. Field experience was seen as an important opportunity to enhance understanding of local health beliefs, health behaviors, and health issues. Students’ learning experiences were gained through observing, reflecting, and sharing, as well as applying what they had learned in the practice of caring for patients. Consequently, some students overcame their fears of the language barrier, different culture, or feeling strange or constrained, and had a positive view of diverse people. “*I understand about culture and can apply my understanding in practice, through communication, when I understand them, I feel more comfortable, and confident when having contact with them. I feel like we were close and there is no longer a distance* (IDSEG1)”.

## 4. Discussion

### 4.1. Development of a Cultural Competence Course

Utilizing the first three phases of the ADDIE model established a systematic process for the development of the cultural competence course. Multi-method needs assessment through conducting a systematic review of cultural competence education and analyzing the viewpoints of nursing educators and students contributed empirical evidence for shaping the cultural competence course. Consequently, course content, educational materials, instructional strategies, duration, time, and evaluation methods that met educational requirements were delivered. The content of the cultural competence course integrated the broad concept of cultural competence generally with cultural care in Vietnam. Previous studies also offered similar course contents within cultural training, including concepts of culture and cultural competence; theoretical models and concepts; and transforming cultural model into nursing practice [9,30,31,32].

The cultural competence course is a combination of lecture and practice within a seven week period, which is moderate in comparison with previous cultural training [33,34,35,36]. Utilizing multiple teaching and learning strategies advocates for the active role of participants and incorporates elements of cultural competence into their cultural care capabilities. Strengthening this approach promotes interaction between educators and learners and deepens understanding of cultural competence through movies and documentaries and learning new cultures. However, the challenge for students was difficulty in writing their reflections.

The formative and summative assessment involved evaluating course validity through a wide range of data from quantitative and qualitative approaches and longitudinal investigation. Formative evaluations were observed during the process of developing the cultural competence course, including nursing instructor and student feedback in the analysis phase, expert assessment of the course blueprint, and direct student feedback after lessons. A summative evaluation was performed using the Kirkpatrick model for training evaluation. To ensure that the process of evaluating the efficacy of the cultural course was valid, instructors and evaluators were different. In sum, this course recognizes the substantial effect of the development of cultural competence in nursing students through an education program.

### 4.2. Evaluation of a Cultural Competence Course

This study set out to evaluate the effect of the cultural competence course on nursing students based on two levels of the Kirkpatrick model. In the baseline phase, the three groups were considered homogenous, having moderate cultural competency scores and low cultural knowledge, skill, and sensitivity. This could be explained due to unfamiliar concepts of cultural competence in Vietnamese nursing education, and participants came from similar backgrounds of education, gender, and social demography. During the first four weeks, all students participated in a common clinical practicum, and cultural competence did not increase and differed between groups. After the training, the findings revealed that the course impacted participants’ overall cultural competence. Student scores on cultural competence increased significantly immediately after completing the course. The most significant change in total cultural competence scores was recognized at the eight-week follow-up. Previous studies have reported similar results over time following their courses [31,33,37]. This cultural training is the result of a combination of wide-ranging approaches among educational models and educational delivery methods to increase engagement during instruction, which benefits all learners and demonstrated positive results. Thus, the findings of this study demonstrate that the cultural competence course is built appropriately to meet the demanding needs of the nursing curricula, particularly in the Vietnamese context.

Educators in the majority of studies opted for a multimodal delivery of the cultural competency curriculum; particularly in the perceived benefits domain. This may suggest that healthcare educators are dedicated to maintaining a wide-ranging approach to increase interest and engagement among students, or that a lack of consensus exists regarding the most effective method, as highlighted by Brottman et al. [27]. The within-subject comparison of improvement between educational modes is an important undertaking to determine whether it is the educational delivery method, rather than the content itself, that determines knowledge retention and attitudinal change.

Mean scores for student cultural knowledge and sensitivity were lower than those for cultural awareness and skill, and there was a significant change in cultural skill before and after the intervention. The findings suggest that students had some awareness of cultural diversity and recognition of their own prejudice, but showed a lack of knowledge of various cultures. After receiving the training, cultural knowledge and sensitivity improved. In particular, cultural skills in cultural health assessment and communication improved to a greater extent than other skills. This positive result was possibly attributable to students’ clinical practice experience in providing care for people from diverse cultures. Additional qualitative data in this study showed that participating in a cultural course is a journey toward cultural competence, not only to gain cultural knowledge and skills but also to change participants’ attitudes. Similar previous studies found that cultural competence improves significantly after taking a cultural course [8,31,38].

Looking to the future, the study findings demonstrate that there is no significant difference between the course plus field experience and the stand-alone course. The reason for this could be linked to the short field experience of only two days’ duration. A similar conclusion was reached by Chen et al. [39], who found that there was no significant difference in cultural competence between the comparison and experimental groups after taking a cultural course following 10 h in-service learning. In contrast, Chang et al. [38] and Stiles et al. [9] revealed that there is a significant positive effect on cultural competence with students who had field experience involving long-term practice of more than four weeks. Additionally, the qualitative result of this study indicates that field experience contributes more to deepening students’ learning experience and cultural values. Similar previous studies demonstrate that field experience contributes to enhancing nursing students’ cultural competence, their community knowledge and experience, and also cultural values [38,40,41].

The reaction level of the Kirkpatrick model showed that students were satisfied with the overall education program. The qualitative analysis indicated that students were very satisfied with the cultural competence course, in terms of course content, teaching and learning approach, and appropriate environment and educational materials. Therefore, this course recognized that both interventions helped students to move toward cultural competence and raised the level of culturally competent nursing care. Various suggestions were recommended for inclusion in a future course; for example, adding more on Vietnamese culture and also explaining the concept of transcultural nursing. These suggestions may prove useful in future cultural competency courses.

The limitation of this study was the application of a convenience sample of undergraduate nursing students at single university enrolled in the nursing program; the sample being limited to females also limits the generalizability of the results. Further research should involve randomization in subject selection and expand to several universities in Vietnam.

## 5. Conclusions and Suggestions

The cultural competence course was constructed based on the ADDIE model, including course structure and course content, and diverse teaching strategies ranging from lecture to field experience promoting students’ cultural competence. The findings of this study support evidence that the incorporation of cultural competency into nursing education curricula enhances the level of cultural competence in undergraduate nursing students. The suggestions expressed for future research regarding cultural competence courses are the following: large class for nursing students; long-term field experience to enhance student cultural competence; development of evaluation tools for checking cultural skill; and training related to cultural competence for nursing educators and nurses should be considered.

## Figures and Tables

**Figure 1 ijerph-19-00888-f001:**
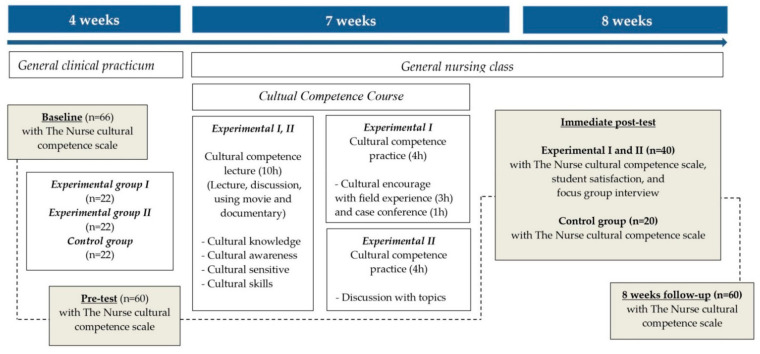
Study flow diagram.

**Figure 2 ijerph-19-00888-f002:**
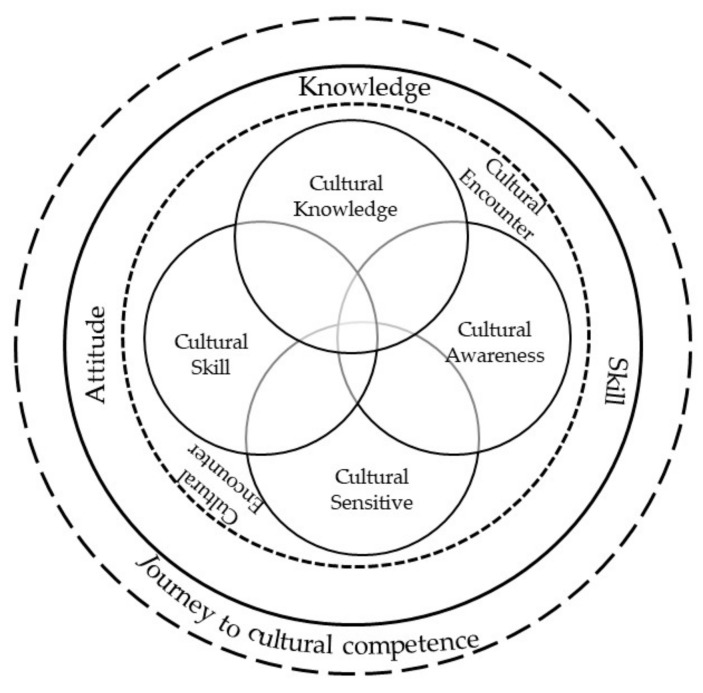
Journey of nursing students to forming cultural competence.

**Table 1 ijerph-19-00888-t001:** Cultural contents for educational training.

Course Contents	Time (h)	Instructional Strategies	Assessment Methods
**Culture** Describe a definition of culture and its impact on related health and illness.	2	Lecture, Watching Movie and documentary, Discussion	Quiz, Reflective writing
**Cultural Diversity** Discuss cultural diversity, its impact on health care.	2		
**Cultural Nursing Competence** Focus on an understanding of cultural competence and effective strategies for promoting culturally competent nursing care.	4		
**Culturally Competent Nursing Care in Vietnamese context** Explain an understanding of culturally competent nursing care in Vietnamese setting.	2		
**Service learning program and case study conference (Experimental group I)** Explain an understanding of culturally competent nursing care Perform cultural assessment by using Purnell model Group reflection and individual reflection about the impact of field experience	4	Service learning, Guest presentation, Home visit Group presentation	Reflective writing, Group report
**Relevant data sources and best evidence in providing culturally competent nursing care (Experimental group II)**	4	Group presentation	Reflective writing, Group report

**Table 2 ijerph-19-00888-t002:** Homogeneity test of general characteristics and NCCS scale among three groups (N = 60).

Variables (N = 60)	Exp. I (*n* = 20)	Exp. II (*n* = 20)	Con. (*n* = 20)	χ^2^/F	*Cramer’s V*	*p*
	Means ± SD/*n* (%)					
Age (years)	20.30 ± 0.57	20.05 ± 0.22	20.40 ± 0.50	2.41		0.10
Religion				3.11	0.16	0.54
Buddhism	1 (5.0)	0 (0.0)	1 (5.0)			
Christian	0 (0.0)	2 (10.0)	2 (10.0)			
No religion	19 (95.0)	18 (90.0)	17 (85.0)			
Take care of race or ethnic patients				1.56	0.16	0.46
Yes	4 (20.0)	5 (25.0)	2 (10.0)			
No	16 (80.0)	15 (75.0)	18 (90.0)			
Live in environment with people with diverse ethnicity				0.17	0.05	0.92
Yes	5 (25.0)	5 (25.0)	6 (30.0)			
No	15 (75.0)	15 (75.0)	14 (70.0)			
Encountered special patients				1.05	0.13	0.59
Yes	13 (65.0)	12 (60.0)	15 (75.0)			
No	7 (35.0)	8 (40.0)	5 (25.0)			
Total NCCS scores	113.65 ± 16.22	113.40 ± 12.11	120.55 ± 10.82	1.88		0.16
Cultural Awareness	37.95 ± 5.32	38.15 ± 4.06	39.20 ± 3.83	0.46		0.64
Cultural Knowledge	20.00 ± 4.97	19.55 ± 4.89	22.40 ± 3.98	2.18		0.12
Cultural Sensitivity	22.75 ± 3.68	23.65 ± 3.69	24.25 ± 3.37	0.89		0.42
Cultural Skills	32.95 ± 7.43	32.05 ± 6.11	34.70 ± 4.46	0.97		0.39

NCCS—Nurse cultural competence scale. Exp.I—Experimental group I; Exp.II—Experimental group II, Con.—Control group.

**Table 3 ijerph-19-00888-t003:** The Comparison of NCCS scale and subscales between baseline and pre-test (N = 60).

Variables (*n* = 60)	Group	T1	T2	Difference (T1–T2)	t	*p*
		Mean ± SD				
Cultural Awareness	Cont.	39.20 ± 3.83	37.55 ± 5.19	1.65 ± 5.15	1.43	0.17
	Exp. I	37.95 ± 5.32	38.00 ± 6.22	−0.05 ± 6.19	−0.04	0.97
	Exp. II	38.15 ± 4.06	35.05 ± 7.16	3.10 ± 7.28	1.90	0.07
Cultural Knowledge	Con.	22.40 ± 3.98	25.15 ± 8.63	−2.75 ± 9.01	−1.36	0.19
	Exp. I	20.00 ± 4.97	20.40± 5.73	−0.40 ± 5.34	−0.34	0.74
	Exp. II	19.55 ± 4.98	19.90 ± 4.32	−0.35 ± 5.31	−0.29	0.77
Cultural Sensitivity	Con.	24.25 ± 3.37	25.35 ± 4.51	−1.10 ± 4.46	−1.10	0.28
	Exp. I	22.75 ± 3.68	24.75 ± 4.69	−2.00 ± 5.05	−1.77	0.09
	Exp. II	23.65 ± 3.69	23.95 ± 5.10	−0.30 ± 5.89	−0.23	0.82
Cultural Skill	Con.	34.70 ± 4.46	36.80 ± 6.59	−2.10 ± 6.50	−1.45	0.17
	Exp. I	32.95 ± 7.43	33.10 ± 7.35	−0.15 ± 8.28	−0.08	0.94
	Exp. II	32.05 ± 6.11	30.95 ± 7.37	1.10 ± 6.25	0.79	0.44
Total NCCS scores	Con.	120.55 ± 10.82	124.85 ± 15.69	−4.30 ± 15.77	−1.22	0.24
	Exp. I	113.65 ± 16.22	116.25 ± 17.69	−2.60 ± 17.09	−0.68	0.52
	Exp. II	113.40 ± 12.11	109.85 ± 16.13	3.55 ± 17.43	0.91	0.37

NCCS—Nurse cultural competence scale. Exp.I—Experimental group I; Exp.II—Experimental group II; Con.—Control group. T1—Baseline; T2—Pre test.

**Table 4 ijerph-19-00888-t004:** The comparison of NCCS score and subscales among three groups (N = 60).

Variables (*n* = 60)	Group	T1	T2	T3	T4	Source	F	*p*
		Mean ± SD						
Cultural Awareness	Cont.	39.20 ± 3.83	37.55 ± 5.19	37.60 ± 5.22	40.00 ± 5.25	Group	0.55	0.582
	Exp. I	37.95 ± 5.32	38.00 ± 6.22	41.70 ± 4.87	41.05 ± 3.93	Time	10.29	<0.001
	Exp. II	38.15 ± 4.06	35.05 ± 7.16	41.65 ± 3.76	41.20 ± 4.01	Time × Group	2.75	0.017
Cultural Knowledge	Con.	22.40 ± 3.98	25.15 ± 8.63	23.50 ± 5.06	23.25 ± 4.18	Group	3.72	0.030
	Exp. I	20.00 ± 4.97	20.40± 5.73	32.50 ± 4.74	32.00 ± 4.00	Time	55.54	<0.001
	Exp. II	19.55 ± 4.98	19.90 ± 4.32	31.15 ± 4.85	31.40 ± 4.64	Time × Group	15.51	<0.001
Cultural Sensitivity	Con.	24.25 ± 3.37	25.35 ± 4.51	25.50 ± 3.99	26.30 ± 3.01	Group	2.89	0.064
	Exp. I	22.75 ± 3.68	24.75 ± 4.69	30.65 ± 3.15	30.15 ± 4.65	Time	25.75	<0.001
	Exp. II	23.65 ± 3.69	23.95 ± 5.10	29.25 ± 4.52	29.05 ± 4.21	Time × Group	3.93	0.002
Cultural Skill	Con.	34.70 ± 4.46	36.80 ± 6.59	35.60 ± 6.00	35.50 ± 5.88	Group	7.87	0.001
	Exp. I	32.95 ± 7.43	33.10 ± 7.35	49.75 ± 7.10	47.10 ± 6.22	Time	64.23	<0.001
	Exp. II	32.05 ± 6.11	30.95 ± 7.37	48.25 ± 6.36	48.20 ± 8.91	Time × Group	17.13	<0.001
Total NCCS scores	Con.	120.55 ± 10.82	124.85 ± 15.69	122.20 ± 12.54	125.05 ± 10.17	Group	7.59	0.001
	Exp. I	113.65 ± 16.22	116.25 ± 17.69	154.60 ± 15.54	150.30 ± 15.74	Time	66.73	<0.001
	Exp. II	113.40 ± 12.11	109.85 ± 16.13	150.30 ± 16.38	149.85 ± 18.68	Time × Group	15.99	<0.001

NCCS—Nurse cultural competence scale. Exp.I—Experimental group I; Exp.II—Experimental group II, Con.—Control group. T1—Baseline; T2—Pre-test; T3—Immediate post-test; T4—8 weeks follow-up.

**Table 5 ijerph-19-00888-t005:** Level of student satisfaction (N = 40).

Variables (*n* = 40)	Total (*n* = 40)	Exp. I (*n* = 20)	Exp. II (*n* = 20)	t	*p*
	Mean ± SD				
Teaching	4.55 ± 0.37	4.62 ± 0.30	4.48 ± 0.42	0.86	0.36
Assessment	3.95 ± 0.54	3.98 ± 0.46	3.92 ± 0.62	2.04	0.16
Generic skills, learning experiences	4.19 ± 0.52	4.38 ± 0.43	4.00 ± 0.48	0.25	0.62
Overall satisfaction	4.40 ± 0.44	4.48 ± 0.45	4.33 ± 0.44	1.89	0.18

Exp.I—Experimental group I; Exp.II—Experimental group II.

**Table 6 ijerph-19-00888-t006:** Qualitative results of effects of cultural competence course.

Category	Subcategory
Nursing student’s perspective on cultural competence course	Journey to cultural competence
	Satisfaction with cultural competence course
	Suggestion for improvements
Nursing student’s reflection of field experience	Obtaining cultural experiences
	Expanding understanding of cultural competence through field experience

## Data Availability

Data are unavailable on request due to privacy and ethical restrictions.

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
