# Peer review of "Development and Evaluation of Cultural Competence Course on Undergraduate Nursing Students in Vietnam"

_ijerph, 2022, doi:10.3390/ijerph19020888_

Round 1

Reviewer 1 Report

I think this manuscript addressed an important issue regarding cultural competence course and evaluating the effects of the course on undergraduate nursing students in Vietnam. The study possesses study significance. The study design was appropriate and in general the paper was well written. The paper has the potential to specifically contribute to our understanding of the aforementioned relationships in a non-western context. I think the manuscript was reasonably organized. However, a few points will need to be further improved. Below list suggestions in my review.

  1. Why did the authors study on Vietnamese students as far as the cultural competence research is concerned?
  2. Please add a section to clarify the theoretical approach in your cultural competence study?
  3. The authors will need to add effect size statistics such as eta and Cramer’s V value whenever the ANOVA and Ch-Square tests were reported. In particular, there were a number of significant differences reported in your Table 3.
  4. The authors have homogeneity test results at the end of p196. In your discussion section, please discuss in details about your entirely non-significant results in your Table 1.
  5. Predominant cultural competence or cross-cultural research indicated that cultural values are important factors in investigating diversely cultural research including your investigating the effects of the course on undergraduate nursing students in Vietnam. I do not however find any cultural value mentions from this study. Furthermore, no relevant citations in this regard were found.
  6. Will the authors also add other main study limitation in addition to simply reporting using a convenience sample procedure?

In sum, this article will need to be further enhanced with greater clarity. The paper needs minor revisions. I am willing to accept this manuscript pending successful revisions on the aforementioned points.

Reviewer 2 Report

Congratulations on the very well documented research! You statistically analysed extensively the data obtained, but I could not see how or where you used atlas.ti. I think your paper could be improved by reporting the qualitive data in a more organized way.

The English could be improved.

Reviewer 3 Report

Dear authors,

First, congratulations for researching an important topic. However, your work has some fundamental shortcomings. These shortcomings are:

In the introduction, the importance of cultural competence education is mentioned. It is valuable in terms of emphasizing the importance of this study. However, what is missing is to explain the concept of cultural competence and to talk about the content of cultural competence education. In the introduction part of the article, the concept of cultural competence should be explained and then the content of cultural competence education should be explained briefly.

In the first paragraph of the method section, the results of the preliminary research and needs analysis applied before the project are explained. However, no evidence is presented. Preliminary research results are based solely on the authors' comments. If this section is based on published research, reference is required, otherwise evidence is required.

In particular, there is a need for a more in-depth presentation of the validity and reliability of qualitative data. What measures were taken to meet the criteria of “credibility, reliability, transferability and verifiability” in qualitative findings? This information must be provided.

Statistical analyzes are not complex enough. Data are currently analyzed using bivariate and nonvariable analyses. A multivariate regression analysis is required that includes all variables in Table 2, Table 3, and Table 4 as independent variables or covariates.

In Table 5, only the qualitative findings as categories and subcategories are presented. Qualitative results can be given in the form of graphs, which are more complex and the relationships with each other are presented through the program mentioned in the study. Comments related to these graphics will have more explanatory and analytical features.

Page 8, line 251. It is necessary to explain what kind of cultural element these groups have other than their individual preferences. These groups can be divided into individuals with different preferences, etc. Although it is possible to name them, it is difficult to qualify them as groups with cultural differences. Explanations and expressions need to be edited. As stated at the beginning of the research, the focus areas of cultural competence and its justifications should be clearly stated.

Section 3.2.1. It would be better if the focus was on the content of the cultural competence training applied rather than the course satisfaction and the related results were presented. Because, although the efficiency of the course is a very special situation for readers and researchers, revealing the positive/negative aspects of the training content -just as the participant with the IDSEG4 code said- will reveal a more generalizable benefit.

Section 4.1. In this section, the development process of the course is explained, but the course content is not mentioned. Explaining the course content will be a good projection for future research and applications. In addition, the comparison with the applications and researches in the literature, which is the most important feature of the discussion section, was insufficient.

Page 10, Lines 333-346. After the training applied in the discussion section, the difference between the groups was emphasized. This is a true and expected statement. However, what is missing is that the authors give information about the possible reasons for the difference between the groups. Instead of just saying "education increases cultural competence", commenting on the reasons will enrich the discussion section.

I wish you success in your work.

Kind regards.

Author Response

Response to Reviewer 3 Comments

Point 1: In the introduction, the importance of cultural competence education is mentioned. It is valuable in terms of emphasizing the importance of this study. However, what is missing is to explain the concept of cultural competence and to talk about the content of cultural competence education. In the introduction part of the article, the concept of cultural competence should be explained and then the content of cultural competence education should be explained briefly

Response 1: In section 2.1 “Development of the Cultural Competence Course”, the concept of cultural competence and contents of this course were added in detail in page 2 lines 54-86.

Point 2: In the first paragraph of the method section, the results of the preliminary research and needs analysis applied before the project are explained. However, no evidence is presented. Preliminary research results are based solely on the authors' comments. If this section is based on published research, reference is required, otherwise evidence is required.

Response 2: Our preliminary research is on the process to public, so we ultilize some refrences in page 2 lines 73-74

“Similaly, previous researches demostrated that it needs to enhance cultural competence for nursing student [3,13,14].”

Point 3:  In particular, there is a need for a more in-depth presentation of the validity and reliability of qualitative data. What measures were taken to meet the criteria of “credibility, reliability, transferability and verifiability” in qualitative findings? This information must be provided.

Response 3: We provide the approaches to meet this point in paper 6 lines 194-202

“Using qualitative content analysis process of Elo and Kyngäs [28] contained three main phases with regard to preparation, organizing, and reporting phase. For pursuit of a trustworthy study, the combination of different methods was served in this study. Interview, observation, and field note was involved for preparation phase; the researches analyzed data carefully follow up on the whole analysis process and categorization for organization phase; and explaining categorization process by using table and use of quotation presented for reporting phase. Additionally, guideline of Elo and colleagues [29] was exploited in order to improving trustworthiness of this study”

Point 4: Statistical analyzes are not complex enough. Data are currently analyzed using bivariate and nonvariable analyses. A multivariate regression analysis is required that includes all variables in Table 2, Table 3, and Table 4 as independent variables or covariates.

Response 4: A multivariate regression analysis consider for analyzing data, however, the result don’t revealve predict factor, except different group. Thus, it isn’t involve in the finally finding.

Point 5: In Table 5, only the qualitative findings as categories and subcategories are presented. Qualitative results can be given in the form of graphs, which are more complex and the relationships with each other are presented through the program mentioned in the study. Comments related to these graphics will have more explanatory and analytical features.

Response 5 As this point, the relationship of part of categories in qualitative findings is described in figure 2. And we revised some s in page 9 lines 267-294.

Point 6: Page 8, line 251. It is necessary to explain what kind of cultural element these groups have other than their individual preferences. These groups can be divided into individuals with different preferences, etc. Although it is possible to name them, it is difficult to qualify them as groups with cultural differences. Explanations and expressions need to be edited. As stated at the beginning of the research, the focus areas of cultural competence and its justifications should be clearly stated.

Response 6: The sentence is revised in paper 9 lines 270-273

 “The course provided sufficient understanding of cultural knowledge, particularly of the culture and health issues of the Vietnamese groups of ethnic as Cotu minority group, lesbian, gay, bisexual, transgender, queer, and others (LGBTQ+) and others”

Point 7: Section 3.2.1. It would be better if the focus was on the content of the cultural competence training applied rather than the course satisfaction and the related results were presented. Because, although the efficiency of the course is a very special situation for readers and researchers, revealing the positive/negative aspects of the training content -just as the participant with the IDSEG4 code said- will reveal a more generalizable benefit.

Response 7: In this study, we ultilized Kirkpatrick model as guide approach to evaluation effect of cultural course in qualitatve and quantitative finding as well, thus the course satisfaction aslo included in this finding. The negative aspect of contents are provided in page 10 lines 305-307 in categories “Suggestion for improvements”.

Point 8: Section 4.1. In this section, the development process of the course is explained, but the course content is not mentioned. Explaining the course content will be a good projection for future research and applications. In addition, the comparison with the applications and researches in the literature, which is the most important feature of the discussion section, was insufficient.

Response 8: In this point, course content was describe in table 1. And the comparison was described in page 10 lines 353-355.

“Previous studies also offerred similar course contents within cultural training including concept of cultural and cultural competence; theoretical models and concepts; and transforming cultural model into nursing practice [9,30–32].”

Point 9: Page 10, Lines 333-346. After the training applied in the discussion section, the difference between the groups was emphasized. This is a true and expected statement. However, what is missing is that the authors give information about the possible reasons for the difference between the groups. Instead of just saying "education increases cultural competence", commenting on the reasons will enrich the discussion section.

Response 9: In page 11 lines 378-380, it explained the reason having differrence between experiemental and control group “This cultual training is results of combination wide-ranging aproaches between educa-tional model and educational delivery method to increase engagement during instruction benefits all learners, this demonstrated positive results. Thus, the findings of this study demonstrate that the cultural competence course is built appropriately to meet the de-manding needs of the nursing curricula, particularly in the Vietnamese context.”

Reviewer 4 Report

The paper is to develop cultural competence course and to evaluate effects of the course on undergraduate nursing students in Vietnam. The topic and the findings of this paper maybe interesing. More experiments are needed to be sure. The latest literature needs to be supplemented.

Author Response

Response to Reviewer 4 Comments

Point 1: The paper is to develop cultural competence course and to evaluate effects of the course on undergraduate nursing students in Vietnam. The topic and the findings of this paper maybe interesing. More experiments are needed to be sure. The latest literature needs to be supplemented.

Response 1: Your comment is very useful for future study. As your feesback, It can be limitation of our study; and need to replicate this course in the future with several univerities.

“The limitation of this study was the application of a convenience sample of undergraduate nursing students at single university enrolled in the nursing program; and the sample being limited to female which limits the generalizability of the results. Further research should involve randomization in subject selection and expand several universities in Vietnam”

Round 2

Reviewer 3 Report

Dear authors,

I think the corrections made by the authors are appropriate and good. The authors did a good job. Congratulations.

I think that one of the points I mentioned for the article is not fully understood. I think it would be appropriate to make a little revision on this issue. "...as Cotu minority group, 311 lesbian, gay, bisexual, transgender, queer, and others (LGBTQ+) and others." I suggested that the explanation be omitted as I think it is actually outside the main topic of this article. Namely, ethnic and local factors can be taken into account as cultural competence/difference, that is, most of them are innate issues apart from individual preferences. But are the different preferences of individuals within the scope of this article and training? This is controversial. If the scope and limitations of the subject can be determined exactly, then it will be clear which subjects will be included in this training and article. The LGBTQ+ section, which is also very limitedly emphasized in the article, can be removed from the article. Apart from this section, the changes I suggested in the article have been made, congratulations.

Kind regards.

Author Response

Dear Reviewer,

I appreciate all great comments for this article which I learn a lot. We are agree with you in this point and are removed it .

We would like to thank again for taking the time to review our manuscript.

Reviewer 4 Report

The purpose of this paper is to develop cultural competence course and to evaluate effects of the course on undergraduate nursing students in Vietnam. The topic is interesting.  A concurrent triangulation mixed-methods study was adopted and the findings support evidence that the incorporation of cultural competency into nursing education curricula enhances the level of cultural competence in undergraduate nursing students.  

Author Response

Dear Reviewer,

We deeply thank your time and efforts that you have dedicated to providing your valuable feedback on my manuscript.